# Combining energy efficiency and quantum advantage in cyclic machines

Waner Hou[1,2,7], Wanchao Yao[1,2,7], Xingyu Zhao ®[1,2,3], Kamran Rehan ®[1,4] ✉,
Yi Li ®[1,2], Yue Li ®[1,2], Eric Lutz ®[5] ✉, Yiheng Lin ®[1,2,3] ✉ & Jiangfeng Du ®[1,2,3,6] ✉

Energy efficiency and quantum advantage are two important features of quantum devices. We here report an experimental realization that combines both features in a quantum engine coupled to a quantum battery that stores the produced work, using a single ion in a linear Paul trap. We begin by establishing the quantum nature of the device by observing nonclassical work oscillations with the number of cycles as verified by energy measurements of the battery. We moreover apply shortcut-to-adiabaticity techniques to suppress quantum friction and improve work production. While the average energy cost of the shortcut protocol is only about 3%, the work output is enhanced by up to approximately 33%, making the machine significantly more energy efficient. We additionally show that the quantum engine consistently outperforms its classical counterpart in this regime. Our results pave the way for energy efficient machines with quantum-enhanced performance.

Future quantum technologies are expected to be energy efficient and offer quantum advantage. A quantum device exhibits quantum advantage when it outperforms its classical counterpart[1], whereas the hallmark of energy efficiency is an improved usage of available energetic resources[2]. Cyclic thermal machines, such as heat engines, have recently been successfully miniaturized to the nanoscale[3–13], where quantum effects, such as coherent superpositions of states[14], are expected to significantly influence their properties at low temperatures. The future development of practical quantum engines faces three critical issues that include (1) the quest of unambiguous observable signatures of quantum behavior[15–17], (2) the suppression of detrimental quantum friction mechanisms[18–20], and (3) the determination of regimes of quantum-enhanced performance[21–23]. Combining the last two aspects is of particular importance in view of limited energy resources, since it would allow to produce more than classically possible at a reduced energetic cost[24]. It, however, requires exquisite control of not only the machine itself, but also of its interaction with the thermal environment, as well as of the coupling to the mechanical output device[25].

Witnessing coherent engine operation is a nontrivial experimental problem[10–12], since measurements usually affect the quantum dynamics of a machine in a negative way. In addition, like classical thermal machines[26], quantum engines are subjected to dissipative losses[18–20]. Quantum friction thus occurs when the Hamiltonian of the working medium of the engine does not commute with that of the external driving[18–20]. Short cycle times, associated with fast driving, lead in this case to nonadiabatic transitions that substantially suppress the work output[18–20]. Lossy quantum devices are not energy efficient, since available resources are not used efficiently. A key task is therefore to design energy efficient machines that deliver more output for similar inputs, without sacrificing power[2].

We here address all these three experimental challenges at the same time. We realize a two-level quantum engine weakly coupled to a quantum harmonic oscillator battery using a single $^{40}$Ca$^+$ ion in a linear Paul trap[27]. The cycle is implemented by driving the ion with a

[1]CAS Key Laboratory of Microscale Magnetic Resonance and School of Physical Sciences, University of Science and Technology of China, Hefei 230026, China. [2]Anhui Province Key Laboratory of Scientific Instrument Development and Application, University of Science and Technology of China, Hefei 230026, China. [3]Hefei National Laboratory, University of Science and Technology of China, Hefei 230088, China. [4]Department of Physics, The University of Haripur, KP, Pakistan. [5]Institute for Theoretical Physics I, University of Stuttgart, D-70550 Stuttgart, Germany. [6]Institute of Quantum Sensing and School of Physics, Zhejiang University, Hangzhou 310027, China. [7]These authors contributed equally: Waner Hou, Wanchao Yao. ✉e-mail: krehan2010@yahoo.com; eric.lutz@itp1.uni-stuttgart.de; yiheng@ustc.edu.cn; djf@ustc.edu.cn

narrow-linewidth laser controlled by an arbitrary waveform generator[28,29], whereas coherent heating and cooling are achieved using laser pumping techniques[27]. We demonstrate the quantum nature of the machine by measuring the energy stored in the battery after a variable number of cycles; the engine is not directly measured in order to preserve its quantum features. The energy of the harmonic oscillator battery does not increase linearly with the cycle number, as for classical engines, but exhibits oscillations that reveal intercycle quantum coherence[16]. We further reduce internal quantum dissipation[18–20] by applying a powerful shortcut-to-adiabaticity protocol[30–39], known as counterdiabatic driving[40–42]. By suppressing nonadiabatic excitations, shortcut-to-adiabaticity methods allow the engineering of adiabatic dynamics in finite time[43–52]. In the following, we extend such techniques to cyclic thermal machines. Whereas the energetic cost of the counterdiabatic driving is only about 3%, we observe an up to 33.2% increase of the work output. Shortcut-to-adiabaticity techniques thus make the engine significantly more energy efficient. Moreover, in this regime, the quantum device produces more work than its classical counterpart, a clear signature of quantum advantage.

## Results

### Model and experimental setup

We consider a quantum engine ($E$) with a qubit as its working medium (Fig. 1). The machine is coupled to a quantum battery ($B$) that consists of a harmonic oscillator that stores the produced work. The corresponding Hamiltonian reads $H = H_E + H_B + H_{EB}$, with $H_E = (\Omega/2) \sigma_x + [v(t)/2]\sigma_z$ and $H_B = \omega a^\dagger a$ (in units of $\hbar$), where $\Omega$ is the frequency of the qubit and $\omega$ that of the oscillator. The operators $\sigma_{x,y,z}$ denote the Pauli matrices, whereas $a$ and $a^\dagger$ are the usual ladder operators of the harmonic oscillator. The engine Hamiltonian $H_E$ is of the Landau-Zener type, a versatile model of a driven two-level system[53], and the function $v(t)$ is the external driving field. Note that the driving term $[v(t)/2]\sigma_z$ does not commute with the Hamiltonian $(\Omega/2)\sigma_x$ of the qubit. The engine-battery coupling is moreover of the form $H_{EB} = -(\eta\Omega/2) \sin(\omega t)\sigma_y(a + a^\dagger)$, with Lamb-Dicke factor $\eta$[27]. We concretely examine a quantum Otto cycle in the diabatic basis of the engine Hamiltonian, that is, in the eigenbasis of $H_E$ with $\Omega = 0$[53], which is composed of the following four steps[54]: (1) Unitary expansion during which the qubit is driven by the field $v(t)$ from $t = 0$ to $\tau/2$, (2) Hot isochore during which the qubit is heated into the excited state in a negligible time, (3) Unitary compression with driving field $v(\tau - t)$ from $t = \tau/2$ to $\tau$, and (4) Cold isochore, which closes the cycle by cooling the qubit back to its ground state in a negligible time (Fig. 1). These four branches correspond to the cycle of a quantum engine coupled to coherent reservoirs[21,22] in the adiabatic basis of the Landau-Zener Hamiltonian, that is, in the instantaneous eigenbasis of $H_E$[53]. Such coherent reservoirs fuel the machine with both heat and

quantum coherence[55,56], in contrast to classical heat engines. We also emphasize that since the cold bath is at zero (positive) temperature and the hot bath at an effective zero (negative) temperature[6], the work produced by the corresponding incoherent engine is the largest possible for a classical two-level Otto cycle with all other parameters fixed.

In our experiment, we store a single $^{40}$Ca$^+$ ion in a linear Paul trap with an ambient magnetic field of 0.538mT, with a motional frequency of $\omega_m = 2\pi \times 2.0$MHz for the axis of interest. The engine qubit is defined by the $D_{5/2}$ ($S_{1/2}$) state that corresponds to the excited $|\uparrow\rangle$ (ground $|\downarrow\rangle$) state, respectively (Supplementary Information). We drive between qubit states with a near resonant quadruple transition by a laser beam of approximately 729 nm, corresponding to a Rabi frequency of $\Omega = 2\pi \times 0.159$MHz. We apply two additional laser beams detuned from the quadruple resonance by approximately $\pm \omega_m$, thus forming the coupling between engine and battery. With proper choice of laser frequencies, a desired Hamiltonian is formed after rotating wave approximation, where the effective harmonic oscillator (with eigenstates $\{|n\rangle\}$) has a frequency of $\omega = 2\pi \times 0.075$MHz, which corresponds to the frequency at which the bichromatic beam is modulated. In the experiment, we prepare the system by first cooling the ion to the Doppler limit by driving the $S_{1/2}$ to $P_{1/2}$ dipole transition. We then apply resolved-sideband cooling on the $S_{1/2}$ to $D_{5/2}$ quadrupole transition to cool the ion motion to the ground state $|n = 0\rangle$ and initialize the qubit to its ground state $|\downarrow\rangle$[57]. Expansion and compression strokes of the quantum Otto cycle in the diabatic basis are implemented with the help of an arbitrary wave generator[28,29] that employs a three-color light field to generate the driving function $v(t)$ with arbitrary amplitude and frequency patterns. We concretely choose $v(t) = v_0(2t/\tau)^2(3 - 4t/\tau)$ with $v_0 = 2\pi \times 0.075$MHz[37]. Heating is realized by optically pumping the ion into the excited state $|\uparrow\rangle$. The work produced during each closed cycle of the engine is stored in the harmonic oscillator, increasing the mean phonon number $\bar{n}$. At the end of the strokes with desired number of cycles, we perform laser thermometry of the battery by mapping the information to the qubit, followed by fluorescence detection. Details of the above processes are available in the Supplementary Information.

### Signature of intercycle quantum coherence

We begin by investigating the quantum features of the device by performing stroboscopic measurements of the energy deposited into the quantum battery[16]. To that end, we determine the (nonadiabatic) mean phonon number $\bar{n}_{NA}(N)$ of the harmonic oscillator after a variable number $N$ of cycles (up to $N = 28$) using standard ion trap techniques[27] (Supplementary Information). We observe periodic oscillations of the work output as a function of $N$ (orange dots in Fig. 2), as predicted in ref. 16, that reveal intercycle quantum coherence. This

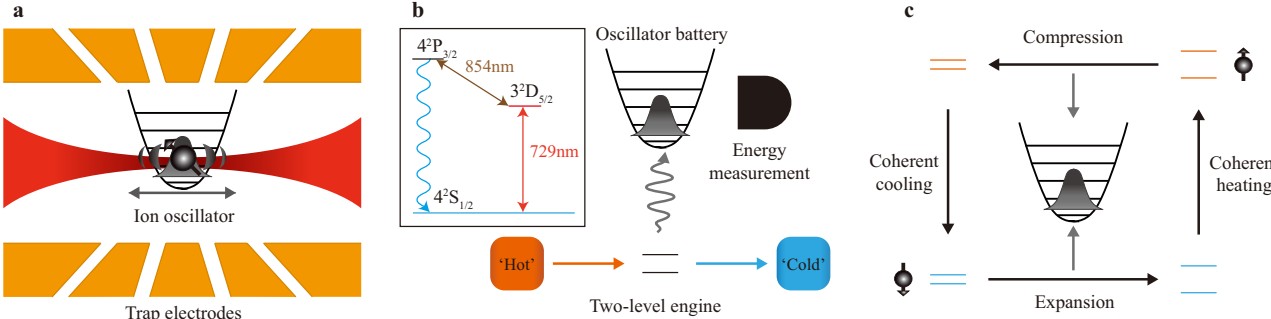

**Fig. 1 | Quantum engine. a** A single ion trapped in a harmonic potential is subjected to control laser fields to realize the quantum machine. **b** A qubit engine cyclically operates between cold and hot coherent baths, and stores the produced work in a quantum harmonic oscillator battery, whose energy is measured after a number of cycles. **c** The cycle consists of four consecutive steps: isochoric expansion, coherent heating, isochoric compression and coherent cooling.

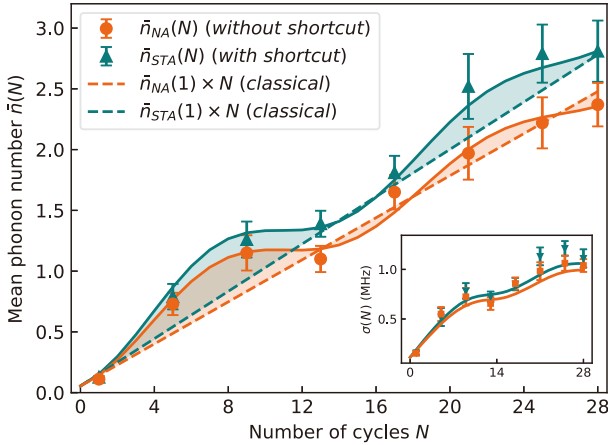

**Fig. 2 | Quantum signature in the engine work output.** The mean phonon number $\bar{n}_{NA}(N)$ determined by an energy measurement of the quantum battery after $N$ cycles exhibits quantum oscillations (orange dots). For a classical engine, the work output scales linearly with $N$ (orange dashed line). With counterdiabatic driving, quantum friction is suppressed and the work output $\bar{n}_{STA}(N)$ is increased (green triangles) above the classical limit (green dashed line). In both cases, good agreement with numerical simulations (solid lines) is obtained. The inset shows the corresponding standard deviations, $\sigma_{NA}(N)$ (orange squares) and $\sigma_{STA}(N)$ (green inverted triangles). Parameters are $v_0 = \omega = 2\pi \times 0.075\,\text{MHz}$ and $\tau = 119\,\mu s$. Error bars correspond to one standard deviation.

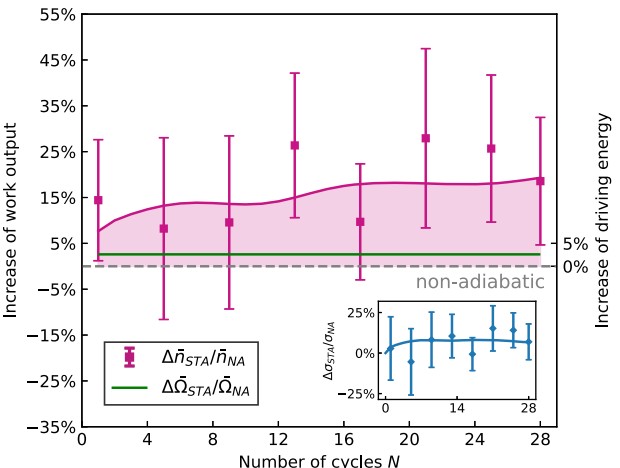

**Fig. 3 | Energy efficient quantum machine.** The relative increase of work output, $\Delta\bar{n}_{STA}/\bar{n}_{NA}$, in the presence of counterdiabatic driving is between 8.2(19.8)% and 27.9(19.5)% (pink squares), depending on the cycle number $N$. By contrast, the average energetic cost of the shortcut protocol, $\Delta\bar{\Omega}_{STA}/\bar{\Omega}_{NA}$, is only about 2.6(0.2)% (green line whose width indicates the experimental error). Available resources are therefore more efficiently used. The inset shows the relative increase of the standard deviation $\Delta\sigma_{STA}/\sigma_{NA}$ (blue diamonds), which is equal to 6.46(14.54)% on average. Solid lines show numerical simulations in both cases. Error bars correspond to one standard deviation.

behavior should be contrasted with that of the corresponding classical machine, where engine and battery always remain diagonal in the energy basis, thus preventing the built-up of intercycle coherence. In this case, according to classical thermodynamics, all engine cycles are identical and independent of each other, implying that the work output scales linearly with the cycle number $N$ (orange dashed line)[16]. We obtain excellent agreement between measured data (orange dots) and numerical simulations (orange line) (Supplementary Information). It should be mentioned that the work output oscillations seen in Fig. 2

cannot be reproduced by driving the oscillator with a classical periodic force, which leads to bounded oscillations of its energy without linear increase[16].

## Counterdiabatic suppression of quantum friction

Another consequence of quantum coherence is quantum friction which occurs when the Hamiltonian of the working medium does not commute with the external driving Hamiltonian[18–20], as is the case in our experiment. We next use a shortcut-to-adiabaticity scheme[41,42] to reduce such quantum dissipation and improve the performance of the quantum machine. As a matter of simplicity and practicality, we do not aim at determining the optimal shortcut protocol, which would require precise knowledge of the open nonunitary dynamics of the qubit engine coupled to the external battery. The needed continuous monitoring of the time evolution would perturb the operation of the quantum machine. We instead add a counterdiabatic Hamiltonian[37,58,59]

$$H_{CD}(t) = -\frac{1}{2}\frac{\Omega\dot{v}(t)}{\Omega^2+v(t)^2}\sigma_y = \frac{\Omega_{CD}(t)}{2}\sigma_y, \qquad (1)$$

with frequency $\Omega_{CD}(t) = -\Omega\dot{v}(t)/[\Omega^2+v(t)^2]$, to the engine Hamiltonian $H_E$ in order to suppress detrimental nonadiabatic transitions[41,42], and reduce coherent oscillations along the $\sigma_y$ direction (Supplementary Information). While Eq. (1) was devised to counter quantum friction for arbitrary driving function $v(t)$[37,58,59], it may also suppress classical friction that could be present. We experimentally realize Eq. (1) by adding another qubit driving component to the three-color light field with a $-\pi/2$ phase difference with respect to $H_E(0)$ to obtain an effective $\sigma_y$ operator that is controlled by the arbitrary wave generator (Supplementary Information).

Figure 2 shows the results of the stroboscopic energy measurement of the quantum battery in the presence of the counterdiabatic driving (1) (green triangles). A robust enhancement of the work output $\bar{n}_{STA}(N)$ is seen, which steadily increases with the number of cycles, up to 20.6(15.2)% for $N = 28$. The counterdiabatic driving hence plays the role of a quantum lubricant[60]. We again obtain excellent agreement between data (green triangles) and numerical simulations (green line). Moreover, in the absence of counterdiabatic driving (orange), the work output oscillates around the classical prediction (orange dashed line). In stark contrast, all data point for the shortcut-driven engine (green) lie above the classical limit with no intercycle coherence (green dashed line). We can therefore conclude that the quantum machine here outperforms its classical counterpart. This represents a distinctive sign of quantum advantage.

## Energy efficiency and cost of the shortcut protocol

Shortcut-to-adiabaticity schemes are not for free as their implementation is usually associated with a given cost. However, this cost is not uniquely defined[42]. We first calculate the average energetic cost of the shortcut protocol during expansion and compression steps by measuring the Rabi frequency $\Omega_{CD}(t)$ of the counterdiabatic driving (1), which is directly accessible in the experiment, and calculating its time average, $\bar{\Omega}_{CD} = \frac{1}{\tau}\int_0^\tau \Omega_{CD}(t)dt$, over one cycle (Supplementary Information). The latter quantity is proportional to the intensity of the laser needed to implement the shortcut Hamiltonian. Compared with the mean Rabi frequency $\bar{\Omega}_{NA}$ of the carrier laser, which is always required to drive the quantum two-level engine, we find a relative increase of $\Delta\bar{\Omega}_{STA}/\bar{\Omega}_{NA} = (\bar{\Omega}_{STA}-\bar{\Omega}_{NA})/\bar{\Omega}_{NA}$ of only 2.6(0.2)% (green line in Fig. 3). This value should be contrasted with the relative increase of work output achieved by the shortcut-to-adiabaticity protocol, $\Delta\bar{n}_{STA}/\bar{n}_{NA} = (\bar{n}_{STA}-\bar{n}_{NA})/\bar{n}_{NA}$, which lies between 8.2(19.8)% and 27.9(19.5)%, with a mean of 17.5(5.8)%, (pink squares in Fig. 3). The quantum lubrication brought about by the counterdiabatic driving hence leads to a significant enhancement of the performance of the engine (of almost one order of magnitude compared to the relatively

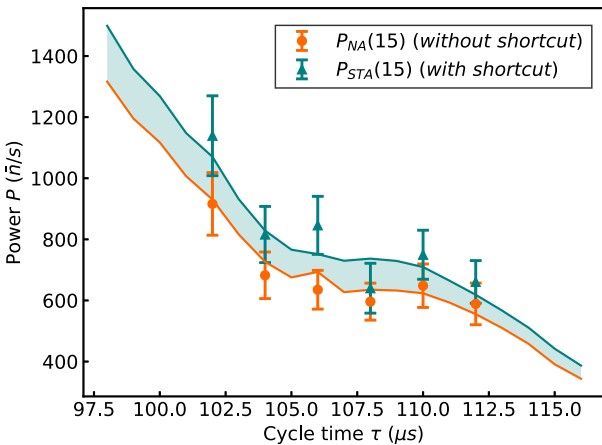

**Fig. 4 | Shortcut enhanced power output.** Power output of the quantum engine, $P = \bar{n}/(N\tau)$, for various cycle times and fixed cycle number $N = 15$ without counterdiabatic driving (orange dots) and with counterdiabatic driving (green triangles). The power output increases when quantum dissipation is suppressed by the shortcut-to-adiabaticity protocol. Good agreement is obtained with numerical simulations (solid lines). Error bars correspond to one standard deviation.

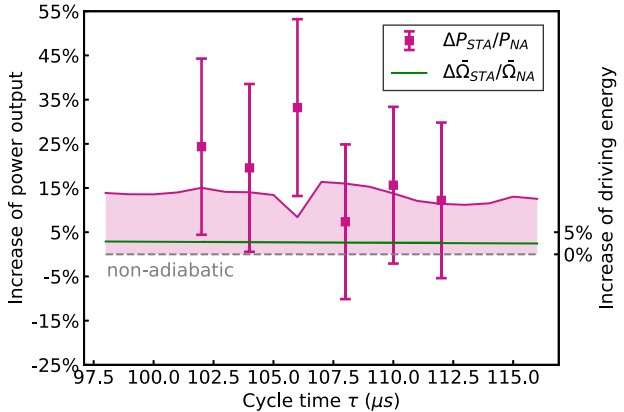

**Fig. 5 | Energy efficient quantum machine.** The relative increase of power output, $\Delta P_{STA}/P_{NA}$, with counterdiabatic driving is between 7.4(17.5)% and 33.2(20.0)% (pink squares), depending on the cycle time $\tau$. By contrast, the average energetic cost of the shortcut protocol, $\Delta\bar{\Omega}_{STA}/\bar{\Omega}_{NA}$, varies between 2.7(0.2)% and 2.9(0.2)% (green line whose width indicates the experimental error). Available resources are thus more efficiently used. Solid lines show numerical simulations. Error bars correspond to one standard deviation.

small energetic cost of the shortcut). This makes the superadiabatic quantum engine more energy efficient, since available resources are exploited much more efficiently after the decrease of quantum friction mechanisms. Note that energy efficiency, which may be defined as the ratio of the minimum energy required to do the job to the energy actually used[2], is related but not identical to the conversion efficiency of the engine, defined as the ratio of work output and heat input[26]. Evaluating the latter quantity would require measuring the heat input, which would negatively affect the quantum features of the engine, and is hence avoided in our experiment.

An additional effect of counterdiabatic driving is to increase the work output fluctuations[61,62]. Another way to assess the cost of the shortcut protocol is thus to evaluate the increase of the standard deviation, $\sigma = (\overline{n^2} - \bar{n}^2)^{1/2}$, seen in Fig. 2 (inset) with (green inverted triangles) and without (orange squares) counterdiabatic driving. We find an average relative increase, $\Delta\sigma_{STA}/\sigma_{NA} = (\sigma_{STA} - \sigma_{NA})/\sigma_{NA}$, of

6.46(14.54)% (Fig. 3, inset), which is again much smaller that the work output enhancement. In fact, since the expectation value of the counterdiabatic Hamiltonian $H_{CD}$ is a factor hundred smaller than the expectation value of the engine Hamiltonian $H_E$ for the parameters of the experiment, the cost of the shortcut is mostly insensitive to the choice of the cost measure (Supplementary Information).

We finally analyze the impact of the cycle time on the cost of the counterdiabatic driving (1). The latter is predicted to increase for shorter cycles, as the dynamics becomes more nonadiabatic[41,42]. Figure 4 displays the power output, $P = \bar{n}/(N\tau)$, of the engine as a function of the cycle time $\tau$ for a constant cycle number $N = 15$ (the background heating rate from the trap has been taken into account and subtracted). We note that both power outputs, without and with shortcut, $P_{NA}$ and $P_{STA}$, strongly increase when the cycle time decreases, as expected. Remarkably, the associated relative power enhancement, $\Delta P_{STA}/P_{NA} = (P_{STA} - P_{NA})/P_{NA}$ (pink squares), presented as a function of $\tau$ in Fig. 5 reaches 33.2(20.0)%, in stark contrast to the associated energetic cost $\Delta\bar{\Omega}_{STA}/\bar{\Omega}_{NA}$ (green line) which is only 2.9(0.2)% on average. Figures 4 and 5 further show that a large increase of the power output does not necessarily imply a large augmentation of the energetic cost, which increases from 2.7(0.2)% for $\tau = 112$ μs to 2.9(0.2)% for $\tau = 102$ μs. For all values of the cycle time, the mean increase of the power output hence largely exceeds the average energetic cost of the counterdiabatic driving, underscoring the improved energy efficiency of the superadiabatic engine.

## Discussion

Assessing the quantum properties of quantum machines is a challenging task, since direct measurements will usually disrupt their delicate quantum features. We have experimentally studied the characteristics of a single ion quantum engine realized in a linear Paul trap by measuring the energy stored in a quantum harmonic oscillator battery. We have observed nonclassical oscillations of the work output with the cycle number, thus revealing intercycle quantum coherence. We have furthermore successfully suppressed quantum friction with the help of shortcut-to-adiabaticity methods that act as a quantum lubricant. In doing so, we have not only increased its performance by about 20% on average, but also significantly increased its energy efficiency, since the net gain is much larger than the thermodynamic cost of the shortcut protocol. We have moreover established its quantum advantage by showing that it consistently outperforms its classical counterpart. Our findings indicate that quantum-enhanced performance can be advantageously combined with energy efficiency. In view of the versatility of our results, which do not depend on the working medium employed or the specific type of engine considered, we expect them to be of importance for the design of energy efficient quantum thermal machines, such as heat engines, refrigerators and pumps[63], as well as of energy efficient quantum technologies[24].

## Data availability

The datasets generated during and/or analysed during the current study are available from the corresponding authors on reasonable request.

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

## Acknowledgements

The USTC team acknowledges support from the National Natural Science Foundation of China (Grant No. 92165206, 11974330), Innovation Program for Quantum Science and Technology (Grant No. 2021ZD0301603), the USTC start-up funding, and the Fundamental Research Funds for the Central Universities, and Hefei Comprehensive National Science Center. K.R. further acknowledges support from Chinese Academy of Sciences President's International Fellowship Initiative (Grant No. 2021PM0049) and China Ministry of Education Funding for Cultivating Key Projects in the Important Directions of Basic Scientific Research (WK3540000004). E.L. is supported by the German Science Foundation (DFG) (Grant No. FOR 2724).

## Author contributions

K.R., E.L., Y. Lin and J.D. conceived the idea and supervised the project. E.L. developed the theory. E.L., Y. Lin, W.H. and W.Y. prepared the manuscript. W.H., X.Z., K.R., Yi Li and Yue Li constructed the experimental apparatus. W.H. collected the data and planned the experiment. W.H. and W.Y. developed the theoretical model, analyzed the data and performed numerical simulations. All authors contributed to the writing of the manuscript.

## Funding

## Competing interests

The authors declare no competing interests.
