## [Transparent Peer Review file · Nature Communications]

Combining energy efficiency and quantum advantage in cyclic machines

Corresponding Author: Professor Eric Lutz

Version 0:

Reviewer comments:

Reviewer #1

(Remarks to the Author)

The current manuscript describes an experimental demonstration of a quantum Otto engine, which is shown to be energy-efficient when run for several cycles. The authors used a single ion in a linear Paul trap and used its electronic states as the working system of this engine. I do not recommend this manuscript for publication in Nature Communications, due to several reasons, as described below:

General issues:

- 1) The manuscript does not present any novel scientific idea and is not expected to advance the field of quantum thermodynamics any further. The idea of using a single trapped ion as a working medium is already known [Chand and Biswas, EPL 118, 60003 (2017), which is not cited though] and the use of counterdiabatic driving to reduce the effect of dissipation is also an established technique [A. del Campo, PRL 111, 100502 (2013)]. In short, this manuscript delivers an amalgam of the essential ideas in an experiment. With due appreciation of the experiment done, I would say that the manuscript is plagued with the absence of novelty.
- 2) Though the author claims in the Conclusions that their results are independent of the working medium used and the type of engine being implemented, they failed to provide any proof that it is really so. Their claim is very generic and emphatic but without any substantial evidence.

Technical issues:

- 1) Why did the authors assume that the bath is coherent? What is the physical nature of this bath and its interaction with the working system?
- 2) The authors did not mention the ambient temperature during the experiment. It seems that the authors did not consider the thermal bath at all. This does not mimic the Otto cycle in any case.
- 3) It is not clear how the hot and cold isochores are performed in a negligible time. Ideally, any heating or cooling takes a very large duration, much larger than the decay time scale of the system. The phrase 'negligible time' does not conform to this fact. Also, this phrase avoids any objectivity in presenting the actual data.
- 4) The energy cost is measured in terms of Ω , which is proportional to the amplitude of the field, not its energy. So, the energy cost could have been calculated in terms of Ω^2 .
- 5) The work is stored into the motional degree of freedom of the ion. The authors claimed that the average phonon number is equivalent to the work stored. Why did they consider it as the work, but not the heat? For higher \bar{n} , the oscillator gets heated. This consideration of \bar{n} as a marker of the work done requires a careful revisit.
- 6) The expression of $v(t)$ is too specific. Is there any reason behind making such a choice of changing $v(t)$ from zero to v_0 ? Is the protocol robust against any other choice of $v(t)$? The authors should have discussed this issue, when they claimed that their method is versatile. The three-color implementation of $v(t)$ should be more explicitly noted in the main body of the

manuscript.

7) An explicit form of $\Omega_{\text{CD}}(t)$ should be noted in the manuscript.

8) What is the decay time-scale of the working system and the oscillator, compared to $N\tau$?

9) An analysis of the efficiency and parameter domain for the heat engine would help in comparison with other works. A plot of efficiency with power or work would identify the optimal domain of energy operation, in support of the Figs. 3 and 5.

In addition to the lack of novelty, there are several technical flaws in the manuscript, which have been overlooked in this manuscript. In view of the above, I do not recommend the acceptance of this manuscript.

Reviewer #2

(Remarks to the Author)

In the manuscript the authors present an experimental implementation of a quantum engine based on a trapped ion platform. They argue that the key features of their engine is that: (i) it allows storage of the extracted energy into a quantum battery (the motion degree of freedom of the ion); (ii) it shows distinctive quantum signatures in its operation; (iii) it takes advantage of being in the quantum regime to outperform its classical counterpart; (iv) and finally they successfully implement counter-adiabatic strategies to improve the engine work output. Overall, I find that the manuscript is technically correct, well-written, and that the authors satisfactorily present evidence of the claims outlined. The topic of experimental implementations of thermodynamics protocols, such as engines, in the quantum regime is highly relevant to a wide audience in the communities of open quantum systems, thermodynamics, and more recently to the fast growing quantum technologies sectors. I particularly find the successful implementation of counter-adiabatic techniques in the engine highly interesting, especially given that the authors show that the energetic gain from it surpasses the cost of implementing it, distinguishing this work from any previous experimental work on engines. It is worth noting here that properly accounting for the cost of implementing counter-adiabatic drivings has attracted a lot of discussion in the literature. While I find the notion put forward by the authors here reasonable, it might be worth having a more extended discussion about the topic somewhere in the manuscript. In conclusion, I overall think that the manuscript is appropriate for publication in Nature Communications, however I first have a few questions and comments for the authors:

1. I find calling the engine a “spin engine” slightly confusing nomenclature. As I understand from the manuscript, the two-level system that plays the role of working medium is not the spin degree of freedom per-se, but the ions $D\ 5/2 - S\ 1/2$ levels. These levels are even called “spin-up” and “spin-down” states. As I said, I find this naming confusing since the electrons in the ion have an actual physical spin degree of freedom (e.g. the internal m_j spin projections within $D\ 5/2$ and $S\ 1/2$) that is not the same as this effective two-level system. I would strongly encourage the authors to reconsider calling these two-levels spin up/down, and to potentially refer to the engine maybe as a “qubit engine” or something else instead of a “spin engine”.

2. I find that Figure 1 is not descriptive enough. In particular, related also to the point above, I believe that it would help greatly to have in Figure 1 a panel with the physical level structure used as the effective two-level system; that is, I would suggest incorporating a version of what is Figure S2 (in the Supplement) into a panel in Figure 1. Moreover, I find the drawing in panel c) of Figure 1 of the two-levels with a circle inside a bit confusing, it is not clear to me what this circle is meant to represent.

3. In the second column of Page 2, the authors state “We concretely examine a quantum Otto cycle in the diabatic base of the engine Hamiltonian, that is, in the eigenbasis of H_E with $\omega = 0$ ”. While I think I understand what authors are trying to say here, I do not think the sentence is clear. Indeed, H_E does not depend on ω at all.

4. In Figure 3 and Figure 5, the caption refers to “green bar” for the energetic cost of the shortcut protocol, however I only see a green line in the plot. Why is it referred to as a green bar? And what exactly would the width of such bar represent? Also, still in Figures 3 and 5, there is a gray shading below the green line that I am not sure what exactly is meant to represent and is not directly referred to anywhere.

5. The authors claim that the counter-adiabatic driving allows them to reduce the effect of quantum friction. It is not fully clear to me the direct evidence of this. I agree that the counter-adiabatic driving provides an advantage in the energy extracted here, but this is also the case in a purely classical situation, where counter-adiabatic driving can also be used. It is therefore not clear to me the connection of the counter-adiabatic driving and quantum-friction.

6. I agree with the authors’ claim of the evidence of quantumness via the non-linear relation in the energy vs cycle number shown in Figure 2. It would have been however quite nice to have direct comparison of this energy to the actual amount of coherence present in the oscillator degree of freedom. Measurement of coherence, and even full state tomography has been successfully done in trapped ion schemes such as the one used here, is there a particular reason this was not possible here?

7. Despite referring to “energy efficiency” several times in the manuscript (including title) somewhat vaguely, I find it somewhat surprising that nowhere in the manuscript they compute the actual thermodynamic efficiency of the engine. Could this be computed? If not, what necessary information do you lack access to, and would it be something that could be feasibly done?

Reviewer #3

(Remarks to the Author)

The authors experimentally study the performance of a coherent quantum engine. Employing a single trapped ion as the working fluid, this work demonstrates that the build up of quantum coherence over many cycles leads to an enhanced performance. This is quantified by coupling the effective two-level system to a harmonic oscillator acting as an energy storage device. This work experimentally verifies some of the theoretical predictions of Ref [16]. The authors also consider the use of quantum control techniques to boost the overall engine performance. Here, by employing a particular metric to quantify the energetic cost of the additional control fields, it is shown that the use of counterdiabatic driving can lead to a significant boost in the power output of the engine.

I believe this work is a useful addition to the field. While the theoretical and experimental techniques employed are already fairly well established, the demonstration of an energetic quantum advantage is in my opinion a significant result for the field. A number of comments I would request the authors to consider:

1) A general comment: overall the work is accessible and well written, however, the discussion is overly terse in many sections. e.g. while substantial space is given to Fig 1, I would have preferred a significantly expanded discussion around the results and techniques. Particularly regarding Figs. 3, 4, and 5, which are all discussed in less than half a page.

Ultimately it was unclear what message/insight the reader was meant to take from these results. The discussion is mostly descriptive. I would invite the authors to significantly expand on the interpretation of these results.

2) A significant result appears to be that the inclusion of the STA enhanced engine leads to a consistent quantum advantage. This is at variance with the unassisted case where the stored energy oscillates around the 'classical' benchmark. While the authors state this in the text, there is no offer of explanation as to why the STA able to consistently outperform the classical limit--can some insight be offered? Is this result specific to how the authors define the cost of the STA? In this regard some more careful discussion on the quantification of the cost of control would help justify this result.

3) A related query regarding the characterisation of the cost: the authors have shown that the average energy of the oscillator/battery indicates a quantum advantage, but I am curious regarding the variance. I would expect that the coherence may impact this. I am aware of alternative measures to quantify the cost of control by considering the variance, e.g. Phys. Rev. Lett. 118, 100602 (2017). I feel some considerations of this point are relevant to strengthening/supporting one of the core conclusions of the work. In this regard also, can the authors comment whether the boosted performance would be maintained if another notion of cost were employed taken from e.g. the various metrics discussed in Rev. Mod. Phys. 91, 045001 (2019).

Reviewer #4

(Remarks to the Author)

In the paper "Combining Energy Efficiency and Quantum Advantage in Cyclic Machines," the authors present an experiment they claim to be a realization of a quantum engine that addresses three crucial experimental challenges: (1) providing unambiguous observable signatures of quantum behavior, (2) suppressing detrimental quantum friction mechanisms, and (3) demonstrating regimes of quantum-enhanced performance. They achieve this using a single trapped ion, where an optical qubit serves as the working medium and the harmonic motion of the ion is used as a "quantum battery" to store the produced work.

As it stands, the manuscript is difficult to follow. First, even as an experienced reader in the field, it is unclear to me where the experiment actually meets claims 1 and 3, due to missing or unclear explanations. Second, I find it challenging to understand some key assumptions and aspects of the correspondence between the model and the physical system. Third, although similar approaches have been discussed in previous work, I question the use of the term "thermal machine" to describe a system that employs coherent heating or cooling. This seems to stretch the definition of "thermal" to the point where it becomes almost oxymoronic. Finally, I struggle to reconcile the concept that the work done by this engine is stored in a battery that simply heats up—this seems more akin to a heat pump than a means of storing work.

Given these issues, I do not believe this paper is currently suitable for publication in Nature Communications. However, I would like to provide some more constructive and detailed criticism to help the authors understand where the text lacks fluidity and clarity:

1. The comparison with the "classical engine" is not adequately described. It is unclear what is meant by a classical engine and how the model is realized. Additionally, if the oscillations are indeed a quantum signature, it should be clearly stated that no classical engine—whether wave-based or oscillatory—could produce such behavior.

2. The presentation of the effective Hamiltonian is very difficult to follow. It is introduced with some parameters not fully described, and then the real Hamiltonian is discussed. I only gained a clear understanding of how the effective Hamiltonian was derived by thoroughly reading the supplementary material. Specifically, the appearance of the frequency ω and its relation to ω_n is not clear. I eventually understood that ω is the frequency at which the bichromatic beam is modulated, but this was only apparent after carefully reading the appendices.

2b. Understanding what ω represents is not a minor detail. It becomes difficult to ascertain whether the battery is in the effective degree of freedom of the coupled Hamiltonian or in that of the free oscillator.

3. You write: "Note that the driving term does not commute with the Hamilton operator of the qubit." What are you referring to as the qubit operator? Is it σ_x ? The driving term is also not defined, but I assume it is the σ_z term. These elements should be clearly defined to avoid confusion.

4. You write: "Hot isochore during which the spin is heated into the excited state in a negligible time." Does this imply a full flip of the population? If so, the spin is not "heated" in the traditional sense, as it will not have a positive temperature

anymore; equal populations correspond to infinite temperature. Please clarify this point.

5. You write: "The states of the quantum engine are defined by the D (...) respectively." Please specify which magnetic sublevels are involved.

6. Please correct the typo in "The quantum lubrication ..." (lubrication).

Version 1:

Reviewer comments:

Reviewer #1

(Remarks to the Author)

The authors have responded to all my questions satisfactorily. I therefore recommend the acceptance of the manuscript in its current form.

Reviewer #2

(Remarks to the Author)

I find that the authors have satisfactorily addressed all the points I previously raised. Moreover, I believe that the revised manuscript has improved overall in terms of clarity and readability. In particular, I appreciate the additional details about the relationship between work oscillations and quantum coherence. Furthermore, I find the new discussion about different metrics to assess the cost of the shortcut protocol and the addition of the quantification of fluctuations very interesting and they contribute to strengthen the claims made by the authors. In conclusion, I recommend the manuscript for publication in Nature Communications.

Reviewer #3

(Remarks to the Author)

The authors have sufficiently addressed all the comments raised in my initial report. In addition, their responses to the other reports are convincing and I believe help demonstrate the importance of the present work. It is my opinion that the present work is suitable for publication in Nature Communications.

Reply to Reviewer #1

We thank the Referee for the detailed report.

“1) The manuscript does not present any novel scientific idea and is not expected to advance the field of quantum thermodynamics any further. The idea of using a single trapped ion as a working medium is already known [Chand and Biswas, EPL 118, 60003 (2017), which is not cited though] and the use of counterdiabatic driving to reduce the effect of dissipation is also an established technique [A. del Campo, PRL 111, 100502 (2013)]. In short, this manuscript delivers an amalgam of the essential ideas in an experiment. With due appreciation of the experiment done, I would say that the manuscript is plagued with the absence of novelty.”

With all due respect, we strongly disagree with the above characterization of our work.

Our manuscript indeed presents a number of experimental ‘firsts’, including (a) the characterization of the quantumness of an engine via stroboscopic energy measurements of the battery, thus verifying the theoretical predictions of Ref. [16] concerning work output oscillations linked to intercycle coherence, (b) the implementation of shortcut-to-adiabaticity techniques to cyclic quantum engines, (c) the evaluation of the energetic cost of the shortcut protocol, (d), the demonstration of energy efficiency associated with a work-output increase much larger than the cost of the shortcut scheme, and (e) the combination of energy efficiency and quantum advantage.

The theoretical paper by Chand and Biswas considers a trapped-ion quantum heat engine that interacts with a hot bath (thermal environment) and a cold bath implemented through the interaction with the external harmonic degree of freedom of the ion. This does not correspond to the model realized in our experiment where the motional degree of freedom corresponds to the flywheel and not to the cold bath. In addition, there is no mention of shortcut-to-adiabaticity methods, quantum signature, quantum advantage or quantum efficiency in that article. Likewise, the theoretical article by del Campo on counterdiabatic driving is neither concerned with shortcut-to-adiabaticity techniques applied to cyclic quantum engines interacting with a battery nor with the issue of evaluating the cost of the shortcut protocol.

We hope we could convince the Reviewer that our manuscript is absolutely not “plagued with the absence of novelty”, but on the contrary corresponds to the state-of-the-art in the field.

The article by del Campo was already cited as Ref. [40]. We have now added the paper by Chand and Biswas as Ref. [4] in the Supplementary Information Sec. I, together with the sentence “*Note that a different ion trap quantum engine, where the motional degree of freedom plays the role of the cold bath, has been discussed in Ref. [4]*”.

“2) Why did the authors assume that the bath is coherent? What is the physical nature of this bath and its interaction with the working system?”

Coherent reservoirs have been theoretically predicted to enhance the performance of engines for more than 20 years, see, for instance, Refs. [21-23]. They are characterized by a density matrix that is thermal along the diagonal, but also exhibit non-diagonal matrix elements (coherence) in the energy basis. Such reservoirs not only exchange heat with the engine, but also quantum coherence, see, for example, Refs. [55,56] (as well as Refs. [21-23]) for detailed models. An important current research topic in experimental quantum thermodynamics is to devise practical schemes to harness quantum coherence and improve engines beyond what is possible classically. As explained in our introduction, this is a highly nontrivial task because of the detrimental effect of decoherence. Our experiment presents for the first time quantum advantage combined with energy efficiency.

We now explain on page 2 that “*These four branches correspond to the cycle of a quantum engine coupled to coherent reservoirs [21-22] in the adiabatic basis of the Landau-Zener Hamiltonian, that is, in the instantaneous eigenbasis of H_E [53]. Such coherent reservoirs fuel the machine with both heat and quantum coherence [55,56], in contrast to classical heat engines*”.

“4) The energy cost is measured in terms of Omega, which is proportional to the amplitude of the field, not its energy. So, the energy cost could have been calculated in terms of Omega².”

We think that this comment is based on a misunderstanding. Equation (1) clearly shows that Ω_{CD} has the dimension of an energy, since H_{CD} is an energy operator (Hamiltonian).

“5) The work is stored into the motional degree of freedom of the ion. The authors claimed that the average phonon number is equivalent to the work stored. Why did they consider it as the work, but not the heat? For higher \bar{n} , the oscillator gets heated. This consideration of \bar{n} as a marker of the work done requires a careful revisit.”

We believe that this comment is based on a confusion. The energy stored in the motional degree of freedom is the work produced by the engine (the engine only exchanges heat with its reservoirs). The concept of a flywheel/battery storing the produced work in a trapped-ion engine has been discussed in detail in the literature in the past, see, for instance, the theoretical article [16] or the two ion-trap experimental realizations of a classical engine [5,8].

“6) The expression of $v(t)$ is too specific. Is there any reason behind making such a choice of changing $v(t)$ from zero to v_0 ? Is the protocol robust against any other choice of $v(t)$? the authors should have discussed this issue, when they claimed that their method is versatile. The three-color implementation of $v(t)$ should be more explicitly noted in the main body of the manuscript.”

Equation (1) shows that the counterdiabatic Hamiltonian can be determined for an arbitrary function $v(t)$. The shortcut protocol is therefore clearly robust against any choice of $v(t)$. We concretely chose the form $v(t) = v_0(2t/\tau)^2(3 - 4t/\tau)$ which was proposed in the theory article [37].

We now indicate below Eq. (1) that the counterdiabatic Hamiltonian “was devised to counter quantum friction for arbitrary driving function $v(t)$ [37,57,58]”.

The implementation of $v(t)$ is described on page 4 below Fig. 4 (as well as in the Supplementary Information).

“7) An explicit form of $\Omega_{CD}(t)$ should be noted in the manuscript.”

An explicit expression of Ω_{CD} was given as the prefactor of σ_y on the left-hand side of Eq. (1). We now specify one more time that it is equal to $\Omega_{CD}(t) = -\Omega\dot{v}(t)/[\Omega^2 + v(t)^2]$ below Eq. (1).

“8) What is the decay time-scale of the working system and the oscillator, compared to $N\tau$?”

The T_2^* time is approximately 5 ms, limited by the heating rate of 240 quanta/s due to noises of the trap electric field, longer than the maximal $N\tau$ of about 3 ms. We verified such an effect by numerical simulation, including all other decoherence effects regarding the phonon and spin degrees of freedom, yet we could still expect coherent oscillations, as shown in Fig. 2 in the main text. These details are now specified in the Supplementary Information Sec. II A.

“9) An analysis of the efficiency and parameter domain for the heat engine would help in comparison with other works. A plot of efficiency with power or work would identify the optimal domain of energy operation, in support of the Figs. 3 and 5.”

The efficiency could be determined in our experiment by evaluating the heat absorbed by the engine (the produced work is already available from measurements of the battery). This would require a full tomography of its density matrix in order to capture both coherent and incoherent contributions to the heat (see, for instance, Phys. Rev. E **102**, 062152 (2020)). However, measuring the engine would perturb its operation and negatively impact its quantum features, which we want to preserve. An important message of our investigation is that it is possible to assess the quantum properties of an engine, as well as the positive effect of shortcut-to-adiabaticity methods, without having to look at the engine itself.

We now mention on page 4 “Note that energy efficiency, which may be defined as the ratio of the minimum energy required to do the job to the energy actually used [2], is related but not identical to the conversion efficiency of the engine, defined as the ratio of work output and heat input [26]. Evaluating the latter quantity would require measuring the heat input, which would negatively affect the quantum features of the engine, and is hence avoided in our experiment”.

Reply to Reviewer #2

We thank the Referee for the detailed and positive report.

“1) I find calling the engine a “spin engine” slightly confusing nomenclature. As I understand from the manuscript, the two-level system that plays the role of working medium is not the spin degree of freedom per-se, but the ions $D 5/2 - S 1/2$ levels. These levels are even called “spin-up” and “spin-down” states. As I said, I find this naming confusing since the electrons in the ion have an actual physical spin degree of freedom (e.g. the internal m_j spin projections within $D 5/2$ and $S 1/2$) that is not the same as this effective two-level system. I would strongly encourage the authors to reconsider calling these two-levels spin up/down, and to potentially refer to the engine maybe as a “qubit engine” or something else instead of a “spin engine”.”

The effective two-level system is commonly referred as ‘spin’ in the trapped-ion community, since it can be exactly mapped onto a spin-1/2 (see, for instance, Sec. III of the review [27]). We however acknowledge that this nomenclature might be confusing for people outside this community. Following the Referee’s suggestion, we have replaced ‘spin engine’ with ‘qubit engine’ throughout the manuscript.

“2) I find that Figure 1 is not descriptive enough. In particular, related also to the point above, I believe that it would help greatly to have in Figure 1 a panel with the physical level structure used as the effective two-level system; that is, I would suggest incorporating a version of what is Figure S2 (in the Supplement) into a panel in Figure 1. Moreover, I find the drawing in panel c) of Figure 1 of the two-levels with a circle inside a bit confusing, it is not clear to me what this circle is mean to represent..”

We have added the physical level structure in Fig. 1 and removed the circles as recommended.

“3) In the second column of Page 2, the authors state “We concretely examine a quantum Otto cycle in the diabatic base of the engine Hamiltonian, that is, in the eigenbasis of H_E with $\omega = 0$ ”. While I think I understand what authors are trying to say here, I do not think the sentence is clear. Indeed, H_E does not depend on ω at all.”

The diabatic basis of the Landau-Zener Hamiltonian is given by the eigenbasis of σ_z [53], that is, of the engine Hamiltonian H_E with $\Omega = 0$ (and not $\omega = 0$). We have corrected the unfortunate typo in the revised version.

“4) In Figure 3 and Figure 5, the caption refers to “green bar” for the energetic cost of the shortcut protocol, however I only see a green line in the plot. Why is it referred to as a green bar? And what exactly would the width of such bar represent? Also, still in Figures 3 and 5, there is a gray shading below the green line that I am not sure what exactly is meant to represent and is not directly referred to anywhere.”

The width of the green bars/lines represents the experimental error. We now refer to them as ‘green lines’ to avoid confusion, and specify that their width *“indicates the experimental error”* in both captions.

“5) The authors claim that the counter-adiabatic driving allows them to reduce the effect of quantum friction. It is not fully clear to me the direct evidence of this. I agree that the counter-adiabatic driving provides an advantage in the energy extracted here, but this is also the case in a purely classical situation, where counter-adiabatic driving can also be used. It is therefore not clear to me the connection of the counter-adiabatic driving and quantum-friction.”

What we can safely claim is that (i) the engine experiences quantum friction because the driving Hamiltonian does not commute with the engine Hamiltonian [18-20] and that (ii) the counterdiabatic driving (1) was devised to counter this quantum friction [37,57,58]. However, the Referee is right: we cannot exclude that the shortcut protocol also suppresses classical friction that may be present.

We now specify below the CD Hamiltonian (1) that *“while Eq. (1) was devised to counter quantum friction [37,57,58], it may also suppress classical friction that could be present.”*

“6) I agree with the authors’ claim of the evidence of quantumness via the non-linear relation in the energy vs cycle number shown in Figure 2. It would have been however quite nice to have direct comparison of this energy to the actual amount of coherence present in the oscillator degree of freedom. Measurement of coherence, and even full state tomography has been successfully done in trapped ion schemes such as the one used here, is there a particular reason this was not possible here? ”

State reconstruction of the oscillator degree of freedom is indeed experimentally possible (see Ref. [27]). However, this is realistically limited to only a small number of levels ($n < 6$) with borderline precision,

Figure S9: Simulated phonon density matrix and its l_1 -norm of coherence (red solid), (a) without and (b) with counterdiabatic driving. The density matrices at the end of 6, 14, 21, 28 cycles are shown; the color and the size of the blocks represent the absolute value of the matrix elements. We also show the numerically simulated mean phonon number, minus the non-oscillating classical contribution, (blue dashed) whose oscillations are directly related to those of l_1 -norm of coherence. Parameters are $\tau = 120 \mu\text{s}$, $\Omega = 2\pi \times 0.159 \text{ MHz}$, $\omega_z = 2\pi \times 2.0338 \text{ MHz}$ and $v_0 = \omega = 2\pi \times 0.075 \text{ MHz}$.

and more levels are populated during the engine cycle. Moreover, errorbars of the average phonon numbers shown in Fig. 2 are around 0.06~0.24, while the values of the simulated off-diagonal components of the density matrices shown above are rather small and around 0~0.1 for typical ones, with rare cases of around 0.2. Thus, the majority of the measurements of the off-diagonal components of the density matrix would be very challenging. However, all the off-diagonal matrix elements are needed to evaluate a coherence measure, such as the l_1 -norm of coherence (see Ref. [14]). There is also the additional problem of the potential instability of the real and imaginary components of the density matrix relative to the external laser drives. Therefore, the full reconstruction of the oscillator density matrix does not seem to be feasible in our experiment at this moment, unfortunately.

However, we agree with the Referee that it would be nice to see an explicit comparison between the energy oscillations and the amount of coherence in the motional degree of freedom. We have thus numerically simulated both quantities using the parameters of the experiment and added the corresponding discussion in the new Sec. IV (“*Work oscillations and quantum coherence*”) of the Supplementary Information. The new Fig. S9 is shown above. We observe a clear connection between the oscillations of the mean phonon number and those of the coherence, as quantified by the l_1 -norm of coherence, both without (a) and with (b) counterdiabatic driving applied to the qubit. The corresponding oscillator density matrices for different values of the cycle number N are displayed below the two plots.

“7) Despite referring to “energy efficiency” several times in the manuscript (including title) somewhat vaguely, I find it somewhat surprising that nowhere in the manuscript they compute the actual thermodynamic efficiency of the engine. Could this be computed? If not, what necessary information do you lack access to, and would it be something that could be feasibly done?”

Energy efficiency may be defined as the “ratio of the minimum energy required to do the job to the en-

ergy actually used”, see, for instance, the APS Energy Efficiency Report [2] on page S8. It is usually nontrivial to evaluate this quantity. It is related, but not identical, to the thermodynamic (conversion) efficiency of a heat engine, which is defined as the ratio of work output and heat input. The latter provides a measure of how good an engine converts heat into work. However, the heat input does not necessarily correspond to the total energy use. A case in point is, for example, when shortcut techniques are employed to reduce friction: it is difficult to argue that the energy needed to implement the shortcut protocol is directly converted into work the way heat is. This extra energy cost is therefore hard to quantify with the usual conversion efficiency, although this has been tried in some theoretical studies [35-38]. Another example is provided by the (conversion) efficiency of a car engine that does not account for the energetic cost required to create the hot bath (that this cost is important becomes clear each time one has to drive to the gas station to refill the tank).

The thermodynamic efficiency of the qubit engine could be determined in our experiment by evaluating the heat absorbed by the engine (the produced work is already available from measurements of the battery). This would require a full tomography of its density matrix in order to capture both coherent and incoherent contributions to the heat (see, for instance, Phys. Rev. E **102**, 062152 (2020)). However, measuring the engine would perturb its operation and negatively impact its quantum features, which we want to preserve. An important message of our investigation is that it is possible to assess the quantum properties of an engine, as well as the positive effect of shortcut methods, without having to look at the engine itself.

We now mention on page 4 “*Note that energy efficiency, which may be defined as the ratio of the minimum energy required to do the job to the energy actually used [2], is related but not identical to the conversion efficiency of the engine, defined as the ratio of work output and heat input [26]. Evaluating the latter quantity would require measuring the heat input, which would negatively affect the quantum features of the engine, and is hence avoided in our experiment*”.

Reply to Reviewer #3

We thank the Referee for the detailed and positive report.

“1) A general comment: overall the work is accessible and well written, however, the discussion is overly terse in many sections. e.g. while substantial space is given to Fig 1, I would have preferred a significantly expanded discussion around the results and techniques. Particularly regarding Figs. 3, 4, and 5, which are all discussed in less than half a page. Ultimately it was unclear what message/insight the reader was meant to take from these results. The discussion is mostly descriptive. I would invite the authors to significantly expand on the interpretation of these results.”

We have reduced Fig. 1 and expanded the discussion of Figs. 3-5 as recommended. We have, in particular, analyzed two different measures of the cost of the counterdiabatic driving: the time averaged Rabi frequency of the shortcut drive and, in a new paragraph on page 4, the standard deviation of the work output – see point 3 below. We have also examined the energetic cost of the shortcut protocol as a function of the cycle number and as a function of the cycle time. The main physical insights of this part are that (i) “*a large increase of the power output does not necessarily imply a large augmentation of the energetic cost*” and (ii) that “*for all values of the cycle time, the mean increase of the power output hence largely exceeds the average energetic cost of the counterdiabatic driving, underscoring the improved energy efficiency of the superadiabatic engine*”.

“2) A significant result appears to be that the inclusion of the STA enhanced engine leads to a consistent quantum advantage. This is at variance with the unassisted case where the stored energy oscillates around the ‘classical’ benchmark. While the authors state this in the text, there is no offer of explanation as to why the STA is able to consistently outperform the classical limit—can some insight be offered? Is this result specific to how the authors define the cost of the STA? In this regard some more careful discussion on the quantification of the cost of control would help justify this result.”

A significant result of our experimental investigation is indeed that STA can lead to a consistent quantum advantage. The fact that the work output with STA oscillates above the classical limit, as seen in Fig. 2, does not depend on how the cost of STA is eventually evaluated (no cost is ‘subtracted’ from the data points). We believe that the important aspect is that, according to the analysis of Ref. [16], such behavior cannot be achieved classically.

To our knowledge the origin of such an effect has not been studied theoretically so far. What we can say is that it depends on the chosen parameters, and that STA does not always provide quantum advantage

(this observation is reminiscent of the result of Ref. [21], where coherence was theoretically found to increase the engine efficiency for some parameters and decrease it for some other parameters). We view this quantum advantage as a tradeoff between two aspects of STA, the increase of the mean value and the increase of the fluctuations. For the data shown in Fig. 2, the increase of the mean is much larger than the increase of the fluctuations, so that oscillations all take place above the classical limit.

“3) A related query regarding the characterisation of the cost: the authors have shown that the average energy of the oscillator/battery indicates a quantum advantage, but I am curious regarding the variance. I would expect that the coherence may impact this. I am aware of alternative measures to quantify the cost of control by considering the variance, e.g. Phys. Rev. Lett. 118, 100602 (2017). I feel some considerations of this point are relevant to strengthening/supporting one of the core conclusions of the work. In this regard also, can the authors comment whether the boosted performance would be maintained if another notion of cost were employed taken from e.g. the various metrics discussed in Rev. Mod. Phys. 91, 045001 (2019).”

We have added a discussion of the standard deviation, as recommend. We have concretely added two insets in Figs. 2 and 3 showing, respectively, the standard deviation of the work output with and without shortcut, σ_{STA} and σ_{NA} , as well as the corresponding relative increase $\Delta\sigma_{STA}/\sigma_{NA} = (\sigma_{STA} - \sigma_{NA})/\sigma_{NA}$. The latter cost is equal to 6.5% on average, a which is again smaller than the work output enhancement. In fact, since the expectation value of the counterdiabatic Hamiltonian H_{CD} is about a factor hundred smaller than the expectation value of the engine Hamiltonian H_E for the parameters of the experiment, the cost of the shortcut is mostly insensitive to the choice of the cost measure. As suggested by the Referee, we have analyzed an alternative metric for the cost of the shortcut protocol discussed in the RMP article: we have concretely numerically compared the time averaged expectation value of the shortcut $(1/\tau) \int_0^\tau \langle H_{CD} \rangle(t) dt$ to that of the engine $(1/\tau) \int_0^\tau \langle H_E \rangle(t) dt$ in Sec. III C (“An alternative cost of the shortcut”) of the Supplementary Information. As before, we refrain from tomographically reconstructing the qubit density matrix during the cycle in order to preserve its quantum properties. We find that the ratio of the two is about 0.4%, which is again a small number. We further mention that the adiabatic work is not accessible in the experiment, since the finite coherence time of the device does not allow us to drive arbitrarily slowly (and still maintain quantum features).

We now indicate on page 4 that “An additional effect of counterdiabatic driving is to increase the work output fluctuations [62,63]. Another way to assess the cost of the shortcut protocol is thus to evaluate the increase of the standard deviation, $\sigma = (\bar{n}^2 - \bar{n}^2)^{1/2}$, seen in Fig. 2 (inset) with (green inverted triangles) and without (orange squares) counterdiabatic driving. We find an average relative increase, $\Delta\sigma_{STA}/\sigma_{NA} = (\sigma_{STA} - \sigma_{NA})/\sigma_{NA}$, of 6.46(14.54)% (Fig. 3, inset), which is again smaller than the work output enhancement. In fact, since the expectation value of the counterdiabatic Hamiltonian H_{CD} is about a factor hundred smaller than the expectation value of the engine Hamiltonian H_E for the parameters of the experiment, the cost of the shortcut is mostly insensitive to the choice of the cost measure (Supplementary Information)”.

Reply to Reviewer #4

We thank the Referee for the detailed report.

“a) First, even as an experienced reader in the field, it is unclear to me where the experiment actually meets claims 1 and 3, due to missing or unclear explanations.”

Both properties are shown in Fig. 2 (‘Quantum signature in the engine work output’) and discussed in Sec. ‘Signature of intercycle quantum coherence’ on page 3 and in Sec. ‘Counterdiabatic suppression of quantum friction’ on page 4.

Claim 1 (‘observable signatures of quantum behavior’) follows from the theoretical analysis performed in Ref. [16] that shows that the work output for a classical engine grows linearly with the cycle number (dashed orange line in Fig. 2). Reference [16] further predicts that a signature of the built-up of quantum coherence between cycles leads to oscillations of the work output which we have observed in our experiment (orange dots in Fig. 2). Additional information about the connection of the oscillations of the work output and the quantum coherence in the energy basis of the harmonic oscillator is now also provided in the new Sec. IV of the Supplementary Information.

Claim 3 (‘demonstration of regimes of quantum-enhanced performance’) can also be seen in Fig. 2. In

the absence of STA (orange), the quantum work output oscillates around the classical prediction (the quantum work output is sometimes larger, sometimes smaller than the classical work output). However, in the presence of STA (green), the quantum work output oscillates above the classical prediction (it is always larger than the classical work output), thus showing quantum advantage.

We have clarified both aspects in the revised manuscript as detailed in point 1 below.

“b) Second, I find it challenging to understand some key assumptions and aspects of the correspondence between the model and the physical system.”

We have addressed this question as described in points 2 and 3 below.

“c) Third, although similar approaches have been discussed in previous work, I question the use of the term “thermal machine” to describe a system that employs coherent heating or cooling. This seems to stretch the definition of “thermal” to the point where it becomes almost oxymoronic.”

Quantum engines that are coupled to coherent heat baths (with both diagonal and off-diagonal density matrix elements in the energy basis) are often referred to as ‘coherent heat engines’ in the literature [21, 22, 55, 56], to reflect the fact that they are fueled by *both* heat and quantum coherence. Although, we find this common nomenclature appropriate, we have followed the Reviewer’s suggestion to avoid any confusion, and only use “*quantum engine*” throughout the revised manuscript.

“d) Finally, I struggle to reconcile the concept that the work done by this engine is stored in a battery that simply heats up – this seems more akin to a heat pump than a means of storing work.”

A heat pump is a device that consumes work to transfer heat from a cold heat bath to a hot heat bath [26]. By contrast, our machine produces work by absorbing energy from a hot coherent bath — and is thus an engine and not a heat pump. The produced work is then used to power a harmonic oscillator — that hence plays the role of a flywheel. This concept of a flywheel/battery storing the produced work has been discussed in detail in the literature in the past, see, for instance, the theoretical article [16] or the two ion-trap experimental realizations of a classical engine [5,8].

“1. The comparison with the “classical engine” is not adequately described. It is unclear what is meant by a classical engine and how the model is realized. Additionally, if the oscillations are indeed a quantum signature, it should be clearly stated that no classical engine—whether wave-based or oscillatory—could produce such behavior.”

The corresponding classical machine is defined by keeping the states of engine and battery diagonal in their energy bases. It can therefore be regarded as the dephased analog of the quantum engine. Consequently, no intercycle coherence can be built up during the operation of the device. According to classical thermodynamics, all cycles of the engine are identical and independent of each other, implying that the work output scales linearly with the number of cycles [16]. This result holds for any classical heat engine type.

We have rewritten the description of Fig. 2 on page 3 to clarify the definition of the corresponding classical engine:

“We observe periodic oscillations of the work output as a function of N (orange dots in Fig. 2), as predicted in Ref. [16], that reveal intercycle quantum coherence. This behavior should be contrasted with that of the corresponding classical machine, where engine and battery always remain diagonal in the energy basis, thus preventing the built-up of intercycle coherence. In this case, according to classical thermodynamics, all engine cycles are identical and independent of each other, implying that the work output scales linearly with the cycle number N (orange dashed line) [16].”

We additionally clarify on page 4 that “*Moreover, in the absence of counterdiabatic driving (orange), the work output oscillates around the classical prediction (orange dashed line). In stark contrast, all data point for the shortcut-driven engine (green) lie above the classical limit with no intercycle coherence (green dashed line)*”.

We furthermore indicated at the bottom of page 3 that “*It should be mentioned that the work output oscillations seen in Fig. 2 cannot be reproduced by driving the oscillator by a classical periodic force, which leads to bounded oscillations of its energy without linear increase [16]*”.

“2. The presentation of the effective Hamiltonian is very difficult to follow. It is introduced with some parameters not fully described, and then the real Hamiltonian is discussed. I only gained a clear understanding of how the effective Hamiltonian was derived by thoroughly reading the supplementary material. Specifically, the appearance of the frequency ω and its relation to ω_n ? is not clear. I eventually understood that ω is the frequency at which the bichromatic beam is modulated, but this was only apparent after carefully reading the appendices.”

We now specify more clearly that “ Ω is the frequency of the qubit and ω that of the harmonic oscillator”. We further mention that “ ω is the frequency at which the bichromatic beam is modulated” as suggested.

“3. You write: “Note that the driving term does not commute with the Hamilton operator of the qubit.” What are you referring to as the qubit operator? Is it σ_x ?? The driving term is also not defined, but I assume it is the σ_z ? term. These elements should be clearly defined to avoid confusion.”

We had previously identified the two terms via their prefactors ($\Omega =$ qubit frequency and $v(t) =$ external driving field). As requested by the Referee, we now write “Note that the driving term ($[v(t)/2]\sigma_z$) does not commute with the Hamilton operator ($(\Omega/2)\sigma_x$) of the qubit”.

“4. You write: “Hot isochore during which the spin is heated into the excited state in a negligible time.” Does this imply a full flip of the population? If so, the spin is not “heated” in the traditional sense, as it will not have a positive temperature anymore; equal populations correspond to infinite temperature. Please clarify this point.”

This indeed corresponds to a full flip and, thus, to an effective zero negative temperature, in line with the experiment reported in Ref. [6]. Following this reference, we also keep the nomenclature ‘heating stroke’. This situation corresponds to the largest possible energy difference (between ground and excited state of the qubit) achievable for a noncoherent heat engine, and, hence, to the largest possible work output. Interestingly, employing coherent reservoirs and shortcut-to-adiabaticity methods allows us to outperform even this negative temperature engine.

We now specify on page 2, after the description of the cycle, that “We also emphasize that since the cold bath is at zero (positive) temperature and the hot bath at an effective zero (negative) temperature [6], the work produced by the corresponding incoherent engine is the largest possible for a classical two-level Otto cycle with all other parameters fixed.”

“5. You write: “The states of the quantum engine are defined by the D (...) respectively.” Please specify which magnetic sublevels are involved.”

The magnetic sublevels were previously indicated in the Supplementary Information, with indicators added in the main text. As recommended, we now specify them in the main text, see the new Fig. 1.

“6. Please correct the typo in “The quantum lubrication ...” (lubrication).”

Corrected, thank you.